# Rescue of Pap-Mas in Systemic JIA Using Janus Kinase Inhibitors, Case Report and Systematic Review

**DOI:** 10.3390/jcm12072702

**Published:** 2023-04-04

**Authors:** Franck Zekre, Anita Duncan, Audrey Laurent, Maud Tusseau, Rémi Pescarmona, Sophie Collardeau-Frachon, Camille Ohlmann, Sébastien Viel, Philippe Reix, Sarah Benezech, Alexandre Belot

**Affiliations:** 1Nephrology, Rheumatology and Dermatology Unit, National Reference Centre for Rare Rheumatic Autoimmune and Systemic Diseases in Children (RAISE), Mère Enfant Hospital, Hospices Civils de Lyon, 69500 Bron, France; 2International Center of Infectiology Research (CIRI), INSERM U1111, CNRS, UMR5308, ENS of Lyon, Claude Bernard University Lyon 1, 69007 Lyon, France; 3Department of Clinical Genetics, Hospices Civils de Lyon, 69500 Lyon, France; 4Immunology Unit, Hôpital Edouard Herriot, 69003 Lyon, France; 5Department of Pathology, Mère-Enfant Hospital, Hospices Civils de Lyon, 69500 Bron, France; 6Service de Pneumologie Pédiatrique, Allergologie et Mucoviscidose, Mère-Enfant Hospital, Hospices Civils de Lyon, 69500 Bron, France; 7Immunology, Allergy and Immunomonitoring Unit, Groupement Hôpitaux du Sud, 69310 Lyon, France; 8UMR CNRS 5558 (équipe EMET), Laboratoire de Biométrie et Biologie Evolutive (LBBE), Université Claude Bernard Lyon 1, 69622 Villeurbanne, France; 9Institute of Hematology and Pediatric Oncology, 69008 Lyon, France

**Keywords:** systemic-juvenile idiopathic arthritis, pulmonary alveolar proteinosis, macrophage activation syndrome, Janus Kinase inhibitors, IL-18

## Abstract

Introduction: Biological disease-modifying anti-rheumatic drugs (bDMARDs) targeting interleukin (IL)-6 and IL-1β represent a steroid-sparing first-line therapy used in systemic-onset juvenile idiopathic arthritis (sJIA). Recently, the occurrence of pulmonary alveolar proteinosis (PAP) in sJIA patients was reported with early-onset and exposure to bDMARDs as potential risk factors. We report on a new case with longitudinal immunomonitoring successfully treated by Janus Kinase inhibitors (JAKi) and review past clinical descriptions of this new entity. Methods: We report one case of pulmonary alveolar proteinosis and macrophage activation syndrome (PAP-MAS) with longitudinal immunomonitoring. We then conducted a review of the literature of seven publications reporting 107 cases of PAP-MAS sJIA, and included the main characteristics and evolution under treatment. Results: Of the seven articles analyzed, the incidence of PAP-MAS among sJIA patients varied from 1.28% to 12.9%. We report here a single case among a cohort of 537 sJIA patients followed in the pediatric department of the Hospices Civils de Lyon over the last 15 years. This child presented with all clinical and immunological characteristics of PAP-MAS. After several lines of treatment, he benefited from JAKi and improved with respect to both systemic symptoms and lung disease. In the literature, strategies with monoclonal antibodies targeting either INF-γ or IL-1β/IL-18 have been tested with variable results. Orally taken JAKi presents the advantage of targeting multiple cytokines and avoiding parenteral injections of monoclonal antibodies that may contribute to the pathogenesis. Conclusions: JAKi represent a promising option in the treatment of lung disease associated with sJIA.

## 1. Introduction

Systemic-onset-juvenile idiopathic arthritis (sJIA) is an autoinflammatory disease occurring in children which is phenotypically close to the adult-onset Still’s disease whose etiology remains unknown. Unlike juvenile idiopathic arthritis (JIA), sJIA is clinically characterized by the presence of prominent systemic signs such as fever, lymphadenopathy, hepatosplenomegaly, arthritis, macular skin rash, and serositis. In combination with all of these clinical disorders, a biological inflammatory syndrome characterized by an elevated C-reactive protein (CRP), hyperleukocytosis, increased serum ferritin and erythrocyte sedimentation rate (ESR) is observed. SJIA is considered an overactivation of the innate immune response, leading to exacerbated production of inflammatory cytokines such as IL-1β, IL-18 and IL-6 [1,2].

Macrophage activation syndrome (MAS) is one of the life-threatening complications occurring in approximately 15–30% patients with sJIA, according to the literature [3,4]. Biologically, MAS is characterized by an excessive proliferation and activation of macrophages and lymphocytes, resulting in the development of a cytokine storm with excessive production of proinflammatory cytokines, in particular IFN-γ. The activation of immune cells and the excessive production of inflammatory cytokines conduct to coagulation disorders with hypofibrinogenemia, elevated liver enzymes, cytopenia in all blood cell lines and hyperferritinemia [5,6].

In recent years, a few observations have reported a lung disease occurring in sJIA patients with MAS, evocative of pulmonary alveolar proteinosis (PAP), a pathology related to a defect in surfactant clearance by macrophages [3,7,8]. Since the initial description in the 2000’s by Kimura et al. [8], different name have been used to define sJIA with MAS and, lung disease. The diagnosis of this complication is based on bronchoalveolar lavage and histopathological examination of lung biopsies highlighting accumulation of macrophages and proteinaceous material in alveolar spaces [9,10].

Radiological abnormalities are found on CT scan, such as ground glass opacities, peri-hilar infiltrates, reticulo-nodular opacities and abnormally low density (−30 to −150 HU) and a “crazy paving” aspect that might suggest the presence of PAP [11].

However, the incidence of PAP in sJIA patients with MAS remains unknown due to the anecdotal nature of reported cases. We first describe here one case that occurred in our center and report the longitudinal immunoprofiling under the consecutive therapeutical strategies and then we discuss the frequency and therapeutical options for PAP in sJIA patients with MAS by a systematic review of the literature.

## 2. Materials and Methods

### 2.1. Patients

Clinical, biological and radiological information was extracted from the patient's medical records. Over a period of 15 years, we observed 537 cases of sJIA and only one case of PAP-MAS in sJIA patients. The study was approved by the Scientific and Ethical Committee of the Hospices Civils de Lyon (23_014). Informed consent was obtained and an information sheet on the use of the data was delivered to the patient and his family.

### 2.2. Review of the Literature

We searched through PubMed databases for articles published between January 2005 and August 2022 using the terms “alveolar proteinosis” and “macrophage activation syndrome in systemic juvenile arthritis”. We selected articles written in English, case reports, and clinical studies, whether prospective, retrospective or designed for clinical trials.

### 2.3. Cytokines Assessment

Plasma concentration of IL-18 was measured by ELLA technology using an enzyme-linked lectin assay instrument (ProteinSimple, BioTechne, Minneapolis, MI, USA).

## 3. Results

### 3.1. Case Report

A young boy was diagnosed with sJIA at the age of 19 months. His case evolved in a remitting/relapsing way with associated symptoms of fever, macular eruption (Figure 1A), polyarthralgia, digital clubbing, adenopathy with a biological inflammatory state (elevated CRP and ESR). He was initially treated with corticosteroids, promptly combined with anti-IL-1 therapy (anakinra). At that time, biological features of macrophage activation syndrome were present and intravenous cyclosporine and pulses of corticosteroids were administered. Initially, corticosteroids and anti-IL-1 partially controlled the disease but the patient relapsed, and several treatments were introduced with moderate efficacy. Anti-IL-6 receptor therapy (tocilizumab) was administered when he was 29 months old, but he developed an anaphylactoid reaction. Canakinumab (anti-IL-1β) and etanercept (anti TNF-α) were also tried without clinical benefit. The sequence of the treatments is summarized in Figure 2.

A chest CT scan performed three years after the onset of the disease in the setting of a persistent cough revealed the presence of a diffuse interstitial syndrome, associating peribronchovascular thickenings and micronodules (Figure 1B). Microbiological screening (PCR and cultures) was negative in blood and bronchoalveolar lavage. Eosinophil counts fluctuated and biological disease-modifying anti-rheumatic drugs (bDMARDs) injections were sometimes associated with hypereosinophilia. The patient carried an HLA-DRB1*15 haplotype. Trio-based whole genome sequencing did not identify pathogenic variants in genes of myeloid/lymphoproliferation with hypereosinophilia. Anti-IL5 mepolizumab was clinically ineffective on the course of disease despite a total disappearance of hypereosinophilia. A chest CT scan performed six months later showed worsening of lung disease (Figure 1C). A lung biopsy was performed, showing accumulation of lipid-filled macrophages, eosinophilic proteinaceous material and cholesterol clefts in alveolar spaces (Figure 1B(1,2)), suggestive of lipid pneumonia/pulmonary alveolar proteinosis. IL-18 was increased above 50,000 pg/mL. All these lesions were compatible with a diffuse lipid pneumonia. Given the pulmonary involvement, the patient was treated with corticosteroids associated with different lines of immunosuppressive drugs (IL-1 and TNF-a blocking drugs). The addition of corticosteroid therapy allowed an improvement of the pulmonary involvement without significant results on the systemic and articular involvement. Indeed, the patient repeated episodes of MAS with persistent joint pain.

For the past two years, the patient has been treated with baricitinib combined with corticosteroid therapy. The addition of baricitinib, a Janus kinase 1 and 2 inhibitor (JAKi) led to a clear improvement in systemic and joint symptoms together with a reduction in inflammation, improvement in respiratory status, disappearance of digital clubbing without recurrence of hypereosinophilia. Eosinophilia at disease onset was around 0.8 G/L. During the MAS episodes, eosinophilia increased at 1 G/L, and culminated during the drug reaction to tocilizumab at 2.59 G/L. After the introduction of JAKi, eosinophil count turned to normal values at 0.22 G/L. With JAKi, we observed a decrease in plasmatic IL-18 levels from 51,686 pg/mL before treatment, to 2405 pg/mL after two years of treatment. Cytokines profiling under treatment by JAKi (baricitinib initially at 4 mg per day and increased to 8 mg per day). In addition, the chest CT scan performed two years after the start of JAKi showed an almost complete disappearance of interstitial syndrome (Figure 1D) and no additional MAS flare was reported (Figure 3).

### 3.2. Literature Review

The PubMed database yielded eight articles. These articles were fully reviewed. We excluded one article [12], because it dealt only with therapeutic approaches in the occurrence of PAP-MAS sJIA without any clinical or biological data.

We finally analyzed seven articles with details on the occurrence of PAP in sJIA. Among these articles, two articles were prospective studies [3,13], one case series of three cases [14], one case report [15], and the other three articles were retrospective single case reports for a total of 107 cases [8,10,16].

Details of the studies are provided in Table 1. The objective of these studies was to determine, the occurrence of lipoid pneumonia in patients with systemic juvenile arthritis. Furthermore, these different studies highlighted the risk factors that may be associated with the onset of alveolar proteinosis.

Based on the available data, the reported incidence of PAP-MAS sJIA is highly variable. Studies with large numbers of patients indicate an frequency between 1.28% and 12.9% [3,8,10].

The time to onset of the disease is between 6 and 18 months. However, in one case report, a patient was consulted for pulmonary symptoms 6 years after the onset of his sJIA [15]. Risk factors include young age at onset, anaphylactoid reactions to tocilizumab, recurrent MAS, elevated IL-18 and in some cases elevation of other non-routine markers such as ICAM 5, eotaxin, MMP7 [16].

The follow-up periods range from 12 and 24 months and the evolution of the disease was reported in only half of the studies, with a mortality rate up to 25%.

The therapeutic strategies were variable in the different studies. They included anti-IL-1 and anti-IL-6 bDMARDs and conventional immunosuppressive drugs. Only two studies [13,14] among the seven articles detailed the treatments received by each patient. In these two studies, two patients were stabilized by a combination of treatments including anti-IL-1 (canakinumab, anakinra) and anti-calcineurins such as cyclosporine and tacrolimus. Five patients did not respond to anti-IL6 (tocilizumab). These patients were in partial remission under anti-IL1, cyclophosphamide and anti-calcineurins association and three of them achieved complete remission with JAKi. One patient was under remission with the use of an anti-INF-γ emapalumab, and finally the use of a JAKi led to an immediate remission in one patient. It is important to underline that one article among these different studies proposed an experimental treatment, MAS 825 [15]. Indeed, this treatment was proposed after an unsatisfactory response to other therapeutic lines. It inhibits the action of IL-1β and IL18. This drug allowed a clinical remission and a normalization of the patient’s biological parameters.

## 4. Discussion

Lung involvement in JIA is rare and probably underestimated, leading some teams to recommend routine checking [17]. Regardless, pulmonary disease associated with JIA is linked to an increased morbidity and mortality [8]. Lung disease can arise in different ways in sJIA; the most common is pleuritis followed by pulmonary arterial hypertension, interstitial lung disease, alveolar proteinosis, fibrosis, lipoid pneumonia. Immunosuppression can also contribute to lung damage, by increasing the risk of infection or direct toxicity of methotrexate [18].

The incidence of sJIA-associated lung disease in these patients is scarcely described in pediatric rheumatology. Here, we report a well-documented observation of a patient who presented with PAP in the context of sJIA accounting for 537 of our local series. This is the only observation out of 537 sJIA patients in our local series. A review of the literature allowed us to collect several patients with the same complication. All the studies mentioned are essentially from North American cohorts.

The young age, in particular the onset of sJIA before the age of 2 years, is associated with a higher threat of sJIA-associated lung disease. The efficacy of IL-1β and IL-6 blockers in the management of sJIA is now well established and these drugs are commonly used in first line treatment for this indication. In parallel, since the beginning of the 2010s, an increase in the occurrence of lipoid pneumonia in patients with sJIA has been reported [10,19]. This increase in the number of cases reported could be due to better diagnostic tools available but could also be related to a larger use of bDMARDs blocking IL-6 and IL-1, as 94% of children with this complication presented after the exposure to at least one biological drug [3]. This “infusion drug reaction with eosinophilia and systemic symptoms-like” reaction was frequently reported and may refer to a specific T cell activation. As in the case of our patient, more than 80% of sJIA patients are carriers of the HLA-DRB1*15 allele, suggesting a role of adaptive immunity into the pathogenesis [20].

The pathophysiological explanation connecting the use of biologicals and the onset of PAP in humans is not yet elucidated. However, in mice models, IL-1 is known to have a key role in the stimulation of pulmonary macrophages [21]. As these macrophages are necessary for the clearance of surfactant [22,23], one can speculate a putative effect of IL-1 blockers on macrophage biology driving an accumulation of surfactant, subsequently promoting the development of PAP.

MAS is characterized by a cytokine storm, responsible for a severe and potentially fatal inflammatory state. This complication affects about 15–30% of patients with sJIA. One of the accepted hypotheses on the occurrence of MAS is related to the involvement of IL-18 in the pathophysiology. This cytokine is known to induce Th1 polarized lymphocytes and to induce the secretion of IFN-γ [13,24]. Furthermore, the production of INF-γ is amplified by another mechanism as the IFN-γ promoter contains a consensus sequence for NF-κB, AP-1. IL-18 activates the IRAK/TRAF6 pathway, which leads to the activation of NF-κB and AP-1. As a result, IL-18 activates INF-γ promoter resulting in synergistic induction of IFN-γ production at the transcriptional level [25]. The cytokine storm caused by IL-18 and IFN-γ production can affect the differentiation of lung macrophages and alter their function and metabolism [26].

The recent description of this complication and the increased use of biologicals in sJIA patients in recent years together with the pseudo-DRESS phenotype, interrogate a possible role of the medication in the development of PAP-MAS [10]. Two hypotheses are proposed by Nigrovic's team [27] to explain the pseudo-DRESS observed in PAP-MAS sJIA patients. The first one states that the presence of HLA-DRB1*15 in sJIA patients promotes an interaction with Th2 polarized CD4 T lymphocytes. The use of anti-IL-6 and IL-6 biomedications, enhance their presentation to Th2 via HLA-DRB1*15. The consequence of this interaction with Th2 is the increased production of eosinophils by secretion of IL-4 and IL-5. The eosinophils will then infiltrate the tissues, resulting in a reaction similar to the DRESS syndrome. The second hypothesis is summarized in the cytokine plasticity hypothesis. In sJIA, there is a significant secretion of IL-1 and IL-6. The consequence is the polarization of TCD4 lymphocytes into Th17 and T reg. The use of treatments blocking IL-1 and IL-6 leads to a switch of CD4 cells to Th1, INF-producing, and Th2, IL4-producing, cells. These polarized cells will recognize antigens presented by HLA-DRB1:15, particularly Th2, resulting in increased eosinophil production and a DRESS-like reaction. Screening of HLA-DRB1*15 might be proposed in the future to drive the DMARDS initial prescription in sJIA patients.

Goldbach Mansky’s team showed that an “IL18 PAP-MAS” entity shared several characteristics with “PAP-MAS sJIA” patients [13]. These characteristics were the occurrence of MAS, blood IL-18 elevation, and the development of lipoid pneumonia. The cytokine IL-18 seems to have a key role in the development of PAP. Indeed, as observed in the study by Rood et al. [15], the initial blockade of IL-1 alone not only did not lead to clinical remission of the patient, but could not prevent the occurrence of pulmonary complications. Treatment targeting IL-18 resulted in a complete cure. In addition, other treatments targeting IL-18 are currently being studied in adult still disease (a condition that shares many of the same characteristics as adult sJIA) [28]. SJIA patients with lung disease have a dramatic increase of circulating IL-18, at a higher level than sJIA without this complication [13].

In addition, patients with IL18 PAP-MAS had a positive IFN signature, reflecting type II IFN exposure. The role of IFN-γ in the occurrence of MAS in sJIA is known and IL-18 acts as an amplifier [8]. In addition, overexpression of IFNγ-regulated genes is found in lung biopsies of patients who developed PAP-MAS sJIA [3]. Therefore, targeting IFN-γ, might be promising [29]. A study investigating the safety and efficacy of Emapalumab is ongoing in the context of MAS-sJIA. Preliminary results show that the administration of emapalumab is effective and safe in controlling SAM [30]. In order to block the cytokine storm, broad-spectrum drugs such as JAKi have been tested and have shown promising results. For example, in 1000 adult patients with rheumatoid arthritis, tofacitinib, a JAK 1 and 3 inhibitors, was more effective than methotrexate and equivalent to adalimumab [31].

The rationale for the use of JAKi in autoimmune diseases is the inhibition of JAK-STAT signaling pathways. These pathways are involved in the initiation and amplification of the inflammatory response. Their direct targeting therefore prevents the production of pro-inflammatory cytokines. In addition, JAKi has the added advantage of being taken per os, unlike anti TNF alpha which requires subcutaneous or intravenous injection. Similarly, Bader-Meunier et al. showed the efficacy and safety of using ruxolitinib (JAK 1 and 2 inhibitor) in a patient treated for sJIA with interstitial lung syndrome [32]. Huang et al., also reported the effective use of tofacitinib in sJIA in a 13-year-old girl [33]. Jorgensen et al. [34] also mentioned the efficacy of baricitinib in a 15-year-old girl treated for sJIA with recurrent MAS and interstitial lung disease. Unlike biotherapies administered subcutaneously or intravenously, JAKi are taken orally. If the route of administration participates to the pathogenesis, oral intake might be more appropriated to block the cytokine storm and avoid the vicious circle, sustained by the drug reaction.

Herein we showed a dramatic improvement of both systemic symptoms and lung disease after stopping biologicals and starting JAKi, even if the global position of JAKi in the global management of JIA associated with lung disease and systemic symptoms requires further study.

## Figures and Tables

**Figure 1 jcm-12-02702-f001:**
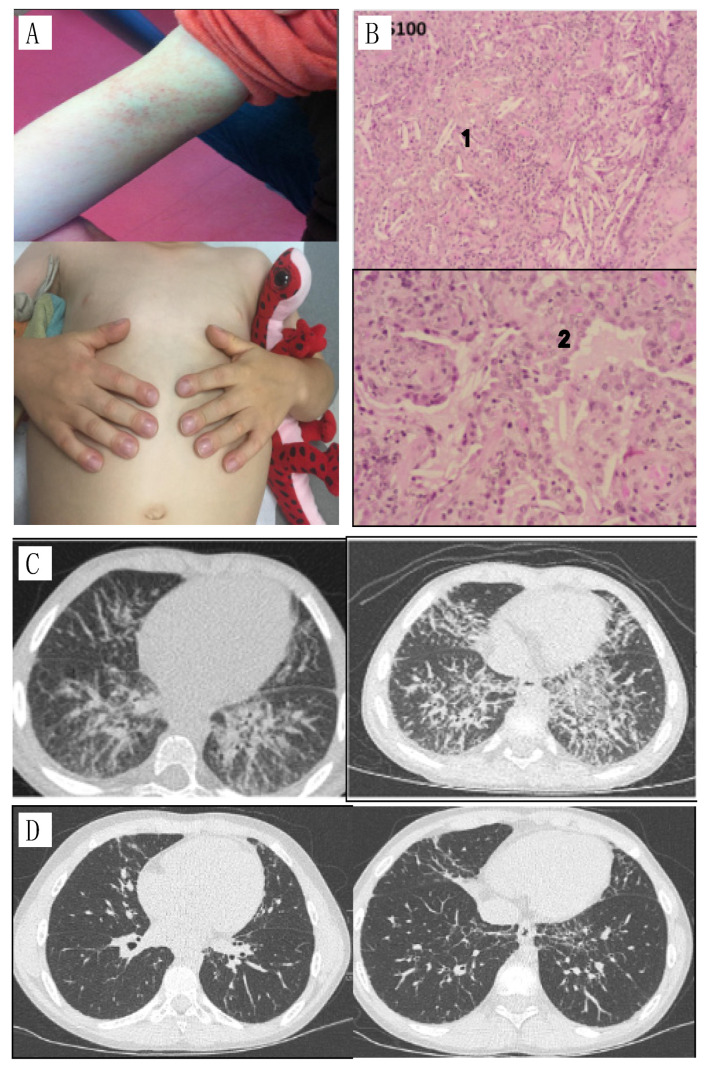
(**A**) specific rash and nail clubbing; (**B**) Lung biopsy, 1: parenchymal crystals, 2: foamy histiocytes; (**C**) CT, diffuse interstitial syndrome, predominantly in the two lower lobes with inter-lobular septa, peribronchovascular thickening and some peribronchovascular and subpleural micronodules. Increased diffuse interstitial syndrome, more extensive in the upper lobes and the anterior territories of the lower and middle lobes; (**D**) CT two years after introduction of baricitinib: significant improvement in bilateral interstitial lung disease.

**Figure 2 jcm-12-02702-f002:**
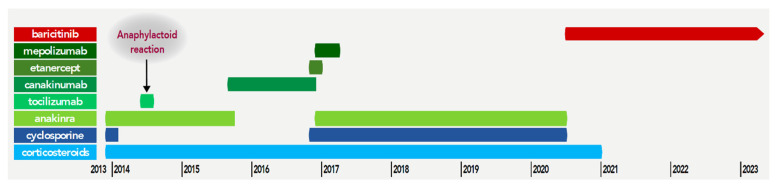
Different lines of treatment.

**Figure 3 jcm-12-02702-f003:**
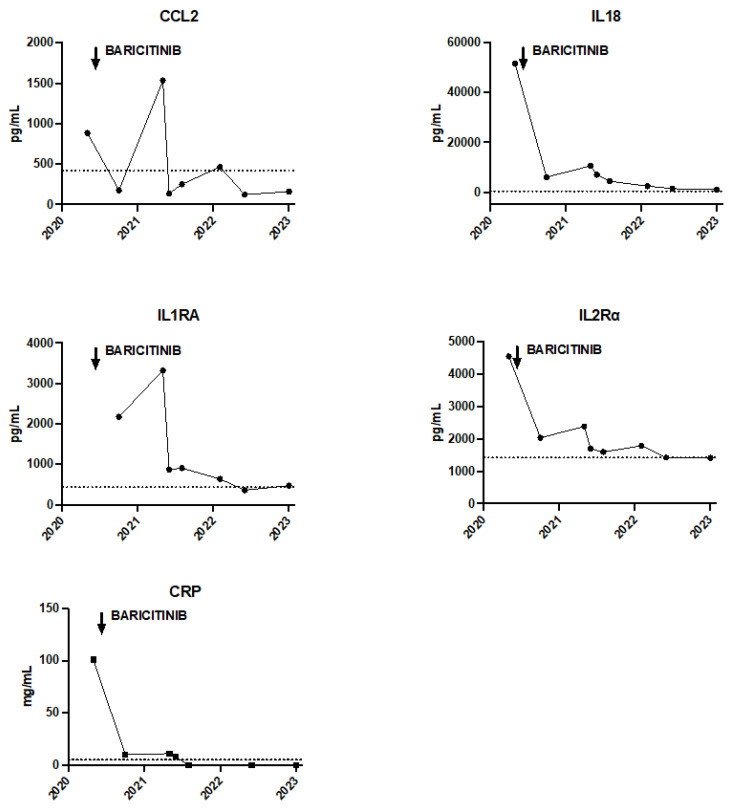
Evolution of the different inflammatory markers with the initiation of Baricitinib. The dotted line corresponds to the upper limit of normal values.

**Table 1 jcm-12-02702-t001:** Studies reporting the occurrence of PAP in sJIA.

Source	Study Design n: Number PAP-sJIA	Frequency PAP-SJIA	Delayed Onset of PAP after the Start of sJIA	Risk Factors	Treatment	Median Follow-Up	Evolution
Schulert (2019) [3]	Prospective study n: 5	6.8%	Not available	Age < 2 years	Anti-IL-1 and anti-IL-6 biologic drugs	12 months	50% improvement associated with disease control
				Tocilizumab reaction			
				Early disease onset MAS			
				IL-18 elevation			
Saper (2019) [10]	Retrospective study n: 61	12.90%	Median time 18 months	Young age	Biological anti cytokine drugs	19 months	Mortality 159/1000 person years
				Anaphylaxis tocilizumab			
				Trisomy 21			
				Onset MAS at the- beginning of the disease			
De Jesus (2020) [13]	Prospective study n: 8	No data	Variable between 6 and 24 months	IL-18 elevation	Anti-IL-1 biologic drugs Ciclosporin Tacrolimus Tofacitinib	24 months	25% death
				IFN positive signature elevation			
				Elevated IgE			
Chen (2022) [16]	Retrospective study n: 24	No data	No data	IL-18 elevation	No data	No data	No data
				ICAM5 elevation			
				MMP7 elevation			
				CCL11 elevation			
				Eotaxin1 elevation			
Canna (2019) [14]	3 cases reports n: 3	No data	12 months	Elevation IL-18	JAK inhibitor Emapalumab	No data	No data
				Age of onset sJIA < 2 years			
				MAS onset			
				Anaphylactic reaction biological drug			
Kimura (2013) [8]	Retrospective study n: 5	1.28%	No data	MAS	DMARD blocking IL1- IL6- TNF alpha Corticoids Cyclophosphamide	No data	No data
Rood (2022) [15]	1 case report n: 1	No data	6 years	IL-18 elevation	Anti IL1 biologic-drugs Methotrexate Cyclophosphamide Azathioprine MAS 825 (biological anti IL1B- anti IL18)	40 weeks	Removal of pulmonary lesions Improvement of clinical signs and complete regression of the inflammatory syndrome
				MAS			

## Data Availability

Not applicable.

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
