# Peer review of "Rescue of Pap-Mas in Systemic JIA Using Janus Kinase Inhibitors, Case Report and Systematic Review"

_jcm, 2023, doi:10.3390/jcm12072702_

Round 1

Reviewer 1 Report

F. Zekre et al reported a case of sJIA complicated by PAP and MAS efficacy treated with JAKi. I believe that the case is interesting and well written, even though the following queries need to be addressed by the authors.

-      I suggest to modify the term SoJIA with sJIA, since sJIA is the same disease both at onset and during disease course.

Introduction

-      The authors reported that “MAS is characterized by …. a cytokine storm with excessive production of proinflammatory cytokines, in particular IFN-γ. In addition, coagulation disorders are observed with hypofibrinogenemia, as well as elevated liver enzymes, cytopenia in all blood cell lines and hyperferritinemia”. Clinical and laboratory features of MAS are a consequence of the cytokine storm not an additional manifestation. I suggest to modify the text accordingly.

Case report

-       The authors reported that “eosinophil counts fluctuated”. I suggest to add eosinophils count, possibly in a table with the laboratory parameters of MAS accordingly with the most relevant disease episode (sJIA onset, full-blown MAS episodes, infusion drug reaction, LD onset, starting JAKi treatment, after 6 mo/1 y after JAKi).

-       I also suggest to add IL-18 levels at disease onset and in different disease points possibly in a table as indicated above, as well as CXCL9 levels in order to evaluate IFNg pathway activation.

-       Did the authors analyze the HLA-DRB1*15 allele in their patient?

-       After the addition of JAKi inhibitor the authors observed an improvement in systemic and joint symptoms and an improvement of respiratory status. I would like also to know if the MAS episodes improved. Did the patient present any other episode after the addition of JAKi?

-       What about the glucocorticoid treatment? Has been never stopped? When compared to the addition of JAKi?

Literature review

-       I suggest to add the recently published case of sJIA-LD treated with Single-Agent Blockade of IL-1β and IL-18 (Rood JE et al. J Clin Immunol. 2023 Jan;43(1):101-108).

-       I would like to know of the authors also evaluated the following case report and if they did why they excluded it from the literature review (Yasin S et al, Rheumatology 2019).

Discussion

-       I suggest to avoid the term DRESS but I would prefer the definition of “infusion drug reaction”.

-       I suggest to modify the following sentence in order to avoid a strong and maybe erroneous message: “Screening of HLA-DRB1*15 might be proposed in the future to drive the DMARDS initial prescription in SoJIA patients”.

-       The study investigating the safety and efficacy of Emapalumab in MAS in the context of sJIA and AOSD is already ended as the authors can verify in the ClinicalTrials.gov identifier NCT 03311854. The preliminary data are already available (De Benedetti F. et al. DOI: 10.1136/annrheumdis-2022-eular.803) and the manuscript is in press.

Table

-       I suggest to improve the quality of the table 1, the data are not clear. Moreover, I suggest to add the number of patients reported in any cited manuscript.

Author Response

Zekre et al reported a case of sJIA complicated by PAP and MAS efficacy treated with JAKi. I believe that the case is interesting and well written, even though the following queries need to be addressed by the authors.

-      I suggest to modify the term SoJIA with sJIA, since sJIA is the same disease both at onset and during disease course.

Thank you for your suggestion, following this comment, we have substituted the term SoJIA by sJIA in the whole manuscript.

Introduction

-      The authors reported that “MAS is characterized by …. a cytokine storm with excessive production of proinflammatory cytokines, in particular IFN-γ. In addition, coagulation disorders are observed with hypofibrinogenemia, as well as elevated liver enzymes, cytopenia in all blood cell lines and hyperferritinemia”. Clinical and laboratory features of MAS are a consequence of the cytokine storm not an additional manifestation. I suggest to modify the text accordingly.

Thank you for your suggestion, we have edited the text as following “The activation of immune cells and the excessive production of inflammatory cytokines conduct to coagulation disorders with hypofibrinogenemia, elevated liver enzymes, cytopenia in all blood cell lines and hyperferritinemia” Line 55-57

Case report

-       The authors reported that “eosinophil counts fluctuated”. I suggest to add eosinophils count, possibly in a table with the laboratory parameters of MAS accordingly with the most relevant disease episode (sJIA onset, full-blown MAS episodes, infusion drug reaction, LD onset, starting JAKi treatment, after 6 mo/1 y after JAKi).

Thank you for your suggestion. We have added in the text the evolution of eosinophils “eosinophilia at the onset of the disease was around 0.8 giga/l. During the MAS episodes eosinophilia increased at 1 giga/l, to culminate during the drug reaction to tocilizumab at 2.59 giga/l. After the introduction of JAKi, eosinophils normalized at 0.22 giga/L”. Line 123-126

-       I also suggest to add IL-18 levels at disease onset and in different disease points possibly in a table as indicated above, as well as CXCL9 levels in order to evaluate IFNgpathway activation.

Thank you for your observation. We added in the manuscript a new figure "figure 3", with additional cytokines that we had for the follow-up of the patient. CXCL9 was not measured but the ISG and plasma level of IFNg remained normal during the whole episode.  The evolution of these parameters after the start of baricitinib is now presented in this figure together with additional cytokines.

-       Did the authors analyze the HLA-DRB1*15 allele in their patient?

Indeed, the patient was a carrier of this allele (mentioned in the line 105). We have now also mentioned this information in the clinical case description. In the other cases treated by JAKi, this information is missing.  

-       After the addition of JAKi inhibitor the authors observed an improvement in systemic and joint symptoms and an improvement of respiratory status. I would like also to know if the MAS episodes improved. Did the patient present any other episode after the addition of JAKi?

Indeed, there were no recurrences of MAS under JAKi treatment. We have clarified this information in the text. Line 129-131

-       What about the glucocorticoid treatment? Has been never stopped? When compared to the addition of JAKi?

As shown in Figure 2, corticosteroids were stopped a few times after the start of JAKi.

Literature review

-       I suggest to add the recently published case of sJIA-LD treated with Single-Agent Blockade of IL-1β and IL-18 (Rood JE et al. J Clin Immunol. 2023 Jan;43(1):101-108).

Thank you for this pertinent suggestion. I was able to add this article published recently in my literature review

-       I would like to know of the authors also evaluated the following case report and if they did why they excluded it from the literature review (Yasin S et al, Rheumatology 2019).

Thank you for your comment, the article by Yasin S et al. does not mention the occurrence of pneumopathy in the patient treated with anti-IL-18. This literature review focus on sJIA patients who have developed the PAP.

Discussion

-       I suggest to avoid the term DRESS but I would prefer the definition of “infusion drug reaction”.

Thank you for your comment. Indeed, it is not a DRESS syndrome in the strict sense of the term, even if this is used in several articles. We wanted to highlight a drug reaction to Tocilizumab injections. In order to avoid any confusion, I have substituted the term DRESS for "infusion drug reaction or pseudo-DRESS" throughout the manuscript. Line 190

-       I suggest to modify the following sentence in order to avoid a strong and maybe erroneous message: “Screening of HLA-DRB1*15 might be proposed in the future to drive the DMARDS initial prescription in SoJIA patients”.

Thank you for the remark, it seems that 80% of "PAP-MAS sJIA" patients carried the HLA-DRB1*15. Other authors and editors have raised this issue of screening and it might be an option for personalized medicine in the future. We have left the sentence as it is.

-       The study investigating the safety and efficacy of Emapalumab in MAS in the context of sJIA and AOSD is already ended as the authors can verify in the ClinicalTrials.gov identifier NCT 03311854. The preliminary data are already available (De Benedetti F. et al. DOI: 10.1136/annrheumdis-2022-eular.803) and the manuscript is in press.

Thanks for the comment: we have made the change on line 243-245

Table

-       I suggest to improve the quality of the table 1, the data are not clear. Moreover, I suggest to add the number of patients reported in any cited manuscript.

Thank you for your pertinent remark. We have now added in the table the number of patients with sJIA-PAP-MAS

Reviewer 2 Report

The manuscript entitled “Rescue of PAP-MAS in SoJIA using Janus Kinase inhibitors, a case report and systematic review” is a case report of an individual with an occurrence of pulmonary alveolar proteinosis (PAP) following a diagnosis of systemic onset juvenile idiopathic arthritis (SoJIA) that was treated with biological disease modifying anti-rheumatic drugs (bDMARDS) targeting IL-6 and IL-1beta and a review of other publications describing other individuals that have developed PAP following a diagnosis of SoJIA that was treated with bDMARDS.

Overall, the paper is interesting and applicable to the state of the field. Strengths of the report are the longitudinal follow-up of one case of SoJIA describing the use of steroids, bDMARDS, and anti-calcineurins, culminating in the success of a Janus Kinase inhibitor to alleviate systemic and pulmonary symptoms and incorporation of this case into the broader scope of other publications describing patients with SoJIA.

Major Comments

1.     On Line 53 – the text indicates that  MAS occurs in approximately 10-15% of patients with SoJIA and cites reference 3.  On Line 186, the text indicates that MAS affects about 30% of patients with SoJIA and cites reference 21. Please clarify why there is a discrepancy between these two publications and which one better reflects the occurrence of MAS in SoJIA patient population.

2.     Figure 1 – Please indicate the age of the patient for each CT scan and/or incorporate the CT images into Figure 2 so the reader can integrate the changes in the CT scans with the changes in the therapies.

3.     Line 211-212 references a study the used a broad spectrum of medications including JAK inhibitors to block cytokine storm. Please include more details about the results of this study.

4.     Please include that Ruxolitinib is a JAK Kinase inhibitor as has been done for etanercept in Line 96.

5.     In table 1, please include the number of patients evaluated in each of the listed publications.

Minor Comments

6.     Line 21 - “is” should be “as” and the “a” should be deleted.  I think the text should read “as potential risk factors”.

7.     Line 24 and Line 52- Please update the sentences to use “macrophage” instead of “macrophagic”

8.     Line 25-26  – “a review of 6 publications reporting cases of …” instead of “6 studies”

9.     Line 26 – “resume” is not the correct word

10.  Line 30 – add the word “with” so the text reads - This child presented “with” all clinical…

11.  Throughout the paper – make sure that IFNg is spelled correctly.

12.  Line 33 – Please update the sentence to says Orally taken JAKi “have” the advantage of ….

13.  Line 34 – Please update the sentence to use “avoiding” instead of “avoid”

14.  Line 76 - Please add the word “case” … observed 537 “cases” of SoJIA…

15.  Line 92 – ciclosporin should be “cyclosporine”

16.  Line 95 – I think it would be helpful to clarify that tocilizumab is an antibody to the IL-6 receptor

17.  Line 96 – I think it would be helpful to add “(anti-IL-1beta)” after Canakinumab.

18.  Line 157 – I don’t think the text “ so should be known of clinicians” is necessary.

19.  Line 206 – Please update the sentence to use “acts” instead of “act”

Author Response

The manuscript entitled “Rescue of PAP-MAS in SoJIA using Janus Kinase inhibitors, a case report and systematic review” is a case report of an individual with an occurrence of pulmonary alveolar proteinosis (PAP) following a diagnosis of systemic onset juvenile idiopathic arthritis (SoJIA) that was treated with biological disease modifying anti-rheumatic drugs (bDMARDS) targeting IL-6 and IL-1beta and a review of other publications describing other individuals that have developed PAP following a diagnosis of SoJIA that was treated with bDMARDS.

Overall, the paper is interesting and applicable to the state of the field. Strengths of the report are the longitudinal follow-up of one case of SoJIA describing the use of steroids, bDMARDS, and anti-calcineurins, culminating in the success of a Janus Kinase inhibitor to alleviate systemic and pulmonary symptoms and incorporation of this case into the broader scope of other publications describing patients with SoJIA.

Major Comments

  1. On Line 53 – the text indicates that MAS occurs in approximately 10-15% of patients with SoJIA and cites reference 3.  On Line 186, the text indicates that MAS affects about 30% of patients with SoJIA and cites reference 21. Please clarify why there is a discrepancy between these two publications and which one better reflects the occurrence of MAS in SoJIA patient population.

Thank you for your relevant comment and we apologize for the inconsistency. Indeed, the incidence is variable across the published papers. We have therefore modified the text and mentioned the incidence from 15 to 30%. (Line 52 and 201)

  1. Figure 1 – Please indicate the age of the patient for each CT scan and/or incorporate the CT images into Figure 2 so the reader can integrate the changes in the CT scans with the changes in the therapies.

Thank you for your comment, we have now added below figure 1 the age of the patient at the time of each CT.

  1. Line 211-212 references a study the used a broad spectrum of medications including JAK inhibitors to block cytokine storm. Please include more details about the results of this study. Thank you for your comment: precision given. Line 245-247. Ref 31

  1. Please include that Ruxolitinib is a JAK Kinase inhibitor as has been done for etanercept in Line 96. We have now mentioned ruxolitinib (JAK 1 and 2 inhibitor) line 293

  1. In table 1, please include the number of patients evaluated in each of the listed publications. Thank you for your pertinent comment. We were able to add the number of patients with sJIA-PAP-MAS

Minor Comments

  1. Line 21 - “is” should be “as” and the “a” should be deleted.  I think the text should read “as potential risk factors”. Changes performed line 21
  2. Line 24 and Line 52- Please update the sentences to use “macrophage” instead of “macrophagic”. Changes performed line 24 and line 51
  3. Line 25-26  – “a review of 6 publications reporting cases of …” instead of “6 studies”. Changes performed line 25-26
  4. Line 26 – “resume” is not the correct word. We replaced resume by "included" line 26
  5. Line 30 – add the word “with” so the text reads - This child presented “with” all clinical… added on line 30
  6. Throughout the paper – make sure that IFNg is spelled correctly. we use “INF-γ» in the manuscript
  7. Line 33 – Please update the sentence to says Orally taken JAKi “have” the advantage of …. Modified as requested line 33
  8. Line 34 – Please update the sentence to use “avoiding” instead of “avoid” Changes done line 33
  9. Line 76 - Please add the word “case” … observed 537 “cases” of SoJIA… Changes done line 77
  10. Line 92 – ciclosporin should be “cyclosporine” appropriate modifications line 93
  11. Line 95 – I think it would be helpful to clarify that tocilizumab is an antibody to the IL-6 receptor Changes performed line 96
  12. Line 96 – I think it would be helpful to add “(anti-IL-1beta)” after Canakinumab. Change performed line 97
  13. Line 157 – I don’t think the text “ so should be known of clinicians” is necessary.

We have removed this sentence accordingly

  1. Line 206 – Please update the sentence to use “acts” instead of “act” Changes performed line 241

Reviewer 3 Report

Zekre and colleagues are presenting a case report of  a child with soJIA PAP-MAS successfully treated with the JAKinib baricitinib, and perform a brief literature review of PAP-MAS/SJIA-LD. Overall this is an interesting contribution to this developing field, and the literature review summarizes the existing clinical studies adequately.

A major concern is the lack of detail concerning the case report's laboratory features (e.g. ferritin levels, CXCL9?, CRP.....) and their development in course of baricitinib treatment, particularly when the goal is to present a "longitudinal immunomonitoring" study- only the levels of IL-18 are mentioned. There is also no mention of dosages for any of the treatments used.

Another concern is, the discussion is lacking in regards to the mechanistics behind jakinibs and how these affect IL18, and the controversy around HLA-DRB1*15 and DRESS vs T cell plasticity hypothesis is not discussed.

Some minor style and text editing is required, some examples are mentioned below.

Detailed line by line review:

Line 22 “we report on a new case”, line 23 same phrase

Line 52, 82 it's called macrophage, not macrophagic activation syndrome

Line 61 could also cite Saper et al, 2019

Line 76 typo/word missing, rephrase sentence

Line 91 dosages of treatments should be mentioned either here or in Figure 2

Line 92 cyclosporine

Fig 1C the text indicates this is 2 CT scans from separate times (line 98, line 106)? If yes, please indicate which panel is first and second CT scan

Fig 1B is described after 1C. It should be re-ordered  so 1B is mentioned in the text before 1C.

Fig 1B would benefit from an indication where specifically lipid filled macrophages, proteinaceous material or cholesterol clefts are observed (e.g. different types of arrows, instead of B1 B2)

As mentioned above as major concern: Are there any other laboratory biomarker levels available for your patient except for IL18 (line 110, line 121)? E.g. CXCL9, CXCL10, Ferritin? Especially if these are available before and after baricitinib therapy, displaying them in a graph together with IL-18 would be highly interesting.

Line 114 “without significant result on systemic and articular involvement”, this is a bit unspecific, can you expand which features didn’t change?

Line 138 time to onset meaning from initial diagnosis of SoJIA?

Line 139 elevated Il-18 is observed in the majority of SoJIA patients, but those developing complications often present with massively increased IL-18 serum levels compared to "just" SJIA

 Line 142 Please specify what you mean by “…and the evolution of the disease was reported in only half of the studies”- how the disease progressed?

149-153 would benefit from indicating in which of the 6 studies these specific patients/treatments are described

157 needs some text/language editing

160/161 citation?

163 this sentence is confusing

175-178 this needs a more expanded description, including on why the DRESS hypothesis is possibly controversial vs T cell plasticity (discussed eg. by Binstadt and Nigrovic https://doi.org/10.1002/art.42137)

185: In the introduction, you stated an MAS occurance rate of 10-15 %, now 30%

187 language editing

189-192 this part seems not particularly important for this discussion

193-194 is this supported by data, or your hypothesis?

195 increased

200-202 I suggest moving this to the introduction

212-214 other studies present case reports of using JAKinibs In sjia / complications, including Huang 2019 PMID: 30948682, Verweyen 2020 PubMed: 31710506 (which also discussed the mechanistics behind JAKinib treatment) and Jorgensen 2020 PMID: 32556329. These should be discussed.

Table 1 needs a different formatting, hard to read with the line breaks

Author Response

Reviewer 3

Zekre and colleagues are presenting a case report of  a child with soJIA PAP-MAS successfully treated with the JAKinib baricitinib, and perform a brief literature review of PAP-MAS/SJIA-LD. Overall this is an interesting contribution to this developing field, and the literature review summarizes the existing clinical studies adequately.

A major concern is the lack of detail concerning the case report's laboratory features (e.g. ferritin levels, CXCL9?, CRP.....) and their development in course of baricitinib treatment, particularly when the goal is to present a "longitudinal immunomonitoring" study- only the levels of IL-18 are mentioned. There is also no mention of dosages for any of the treatments used.

Thank you for your observation. We added in the manuscript a new figure "figure 3", with all the inflammatory parameters that we had for the follow-up of the patient. The evolution of these parameters after the start of baricitinib is mentioned in this figure.

Regarding the dosage of the drugs, we added the doses of baricitinib used during follow-up.

Another concern is, the discussion is lacking in regards to the mechanistics behind jakinibs and how these affect IL18, and the controversy around HLA-DRB1*15 and DRESS vs T cell plasticity hypothesis is not discussed. Thank you for the suggestion. We have now discussed this point in the manuscript. Line 213-226

Some minor style and text editing is required, some examples are mentioned below.

Detailed line by line review:

Line 22 “we report on a new case”, line 23 same phrase. Thank you for this comment, we rephrase the sentence to avoid repetition line 22 et 23.

Line 52, 82 it's called macrophage, not macrophagic activation syndrome. changes made in all text

Line 61 could also cite Saper et al, 2019. Quote added. Line 65

Line 76 typo/word missing, rephrase sentence. Word " cases " added line 77

Line 91 dosages of treatments should be mentioned either here or in Figure 2. Regarding the dosage of the drugs, we have carried out different dosages during the follow-up and we have added the doses of baricitinib used during follow-up.

Line 92 cyclosporine. Adjustment made. Line 93

Fig 1C the text indicates this is 2 CT scans from separate times (line 98, line 106)? If yes, please indicate which panel is first and second CT scan. Yes. Clarification added to figure 1

Fig 1B is described after 1C. It should be re-ordered so 1B is mentioned in the text before 1C. Reorganization in coherence with the order of quotation made. Figure 1

Fig 1B would benefit from an indication where specifically lipid filled macrophages, proteinaceous material or cholesterol clefts are observed (e.g. different types of arrows, instead of B1 B2). Thank you for this suggestion, we have clarified the legend for the figure (now C1, C2)

As mentioned above as major concern: Are there any other laboratory biomarker levels available for your patient except for IL18 (line 110, line 121)? E.g. CXCL9, CXCL10, Ferritin? Especially if these are available before and after baricitinib therapy, displaying them in a graph together with IL-18 would be highly interesting. Thank you for your observation. We added in the manuscript a new figure "figure 3", with all the inflammatory parameters that we had for the follow-up of the patient. The evolution of these parameters after the start of baricitinib is now mentioned in this figure.

 Line 114 “without significant result on systemic and articular involvement”, this is a bit unspecific, can you expand which features didn’t change? Clarification added. “Indeed, the patient relapsed his episodes of MAS with persistent joint pain”. Line 116-118

Line 138 time to onset meaning from initial diagnosis of SoJIA? Yes, the time between diagnosis sJIA and the occurrence of PAP. It has been clarified.

Line 139 elevated Il-18 is observed in the majority of SoJIA patients, but those developing complications often present with massively increased IL-18 serum levels compared to "just" SJIA

We agree with this comment and now underline the point of the major increase of IL-18 in sJIA with LD in the discussion.

 Line 142 Please specify what you mean by “…and the evolution of the disease was reported in only half of the studies”- how the disease progressed?
In Table 1, the progression of the disease was available in 4 studies out of 7.

 149-153 would benefit from indicating in which of the 6 studies these specific patients/treatments are described. Thank you for your comment, however, few studies report on the treatments received in detail. The different studies reported globally the main lines of treatment with failures and successes of immunosuppressants.

 157 needs some text/language editing. review performed line 170-171

160/161 citation ? Ref 18 « Jakubovic et al… »

163 this sentence is confusing. Reformulation performed. Line 177-178

175-178 this needs a more expanded description, including on why the DRESS hypothesis is possibly controversial vs T cell plasticity (discussed eg. by Binstadt and Nigrovic https://doi.org/10.1002/art.42137).

We have added a paragraph on cytokine plasticity to the discussion. Line 213-226

185: In the introduction, you stated an MAS occurrence rate of 10-15 %, now 30%

Thank you for your relevant comment and we apologize for the inconsistency. Indeed, the incidence is variable across the published papers. We have therefore modified the text and mentioned the incidence from 15 to 30%. (Line 52 and 201)

187 language editing. Editing line 203-204.

189-192 this part seems not particularly important for this discussion.

Thank you for your feedback but we wanted here to highlight that IL18 can promote and amplify the occurrence of MAS. Ref 13, Ref 24

193-194 is this supported by data, or your hypothesis?

Thank you for your comment. Indeed, this refers to published work, mentioned in reference 26 (eBIOMedecine 2018)

195 increased review performed line 211

200-202 I suggest moving this to the introduction.

thank you for your suggestion. We have moved the sentence to the introduction

212-214 other studies present case reports of using JAKinibs In sjia / complications, including Huang 2019 PMID: 30948682, Verweyen 2020 PubMed: 31710506 (which also discussed the mechanistics behind JAKinib treatment) and Jorgensen 2020 PMID: 32556329. These should be discussed.

Thank you for your very pertinent suggestion, indeed we have discussed those manuscripts in which JAKi have been used.

Table 1 needs a different formatting, hard to read with the line breaks. Reformatted table 1

Round 2

Reviewer 3 Report

Overall the revision has greatly improved this manuscript. Very good to include the new Fig 3. The discussion is now more comprehensive.